# Study on the Neuroprotective, Radical-Scavenging and MAO-B Inhibiting Properties of New Benzimidazole Arylhydrazones as Potential Multi-Target Drugs for the Treatment of Parkinson’s Disease

**DOI:** 10.3390/antiox11050884

**Published:** 2022-04-29

**Authors:** Neda Anastassova, Denitsa Aluani, Nadya Hristova-Avakumova, Virginia Tzankova, Magdalena Kondeva-Burdina, Miroslav Rangelov, Nadezhda Todorova, Denitsa Yancheva

**Affiliations:** 1Institute of Organic Chemistry with Centre of Phytochemistry, Bulgarian Academy of Sciences, Acad. G. Bonchev Str., Building 9, 1113 Sofia, Bulgaria; miroslav.rangelov@orgchm.bas.bg (M.R.); denitsa.pantaleeva@orgchm.bas.bg (D.Y.); 2Laboratory of Drug Metabolism and Drug Toxicity, Department of Pharmacology, Pharmacotherapy and Toxicology, Faculty of Pharmacy, Medical University-Sofia, 2 Dunav Str., 1000 Sofia, Bulgaria; daluani@pharmfac.mu-sofia.bg (D.A.); vtzankova@pharmfac.mu-sofia.bg (V.T.); magdalenakondeva@abv.bg (M.K.-B.); 3Department of Medical Physics and Biophysics, Faculty of Medicine, Medical University of Sofia, 2 Zdrave Str., 1431 Sofia, Bulgaria; nadia_hristova@abv.bg; 4Institute of Biodiversity and Ecosystem Research, Bulgarian Academy of Sciences, 2 Gagarin Str., 1113 Sofia, Bulgaria; nadeshda@abv.bg

**Keywords:** Parkinson’s disease, MAO-B inhibition, neuroprotection, synaptosomes, benzimidazoles, MTDLs, synthesis, superoxide scavenging, lipid peroxidation, catechols

## Abstract

Oxidative stress is a key contributing factor in the complex degenerating cascade in Parkinson’s disease. The inhibition of MAO-B affords higher dopamine bioavailability and stops ROS formation. The incorporation of hydroxy and methoxy groups in the arylhydrazone moiety of a new series of 1,3-disubstituted benzimidazole-2-thiones could increase the neuroprotective activity. In vitro safety evaluation on SH-SY5Y cells and rat brain synaptosomes showed a strong safety profile. Antioxidant and neuroprotective effects were evaluated in H_2_O_2_-induced oxidative stress on SH-SY5Y cells and in a model of 6-OHDA-induced neurotoxicity in rat brain synaptosomes, where the dihydroxy compounds **3h** and **3i** demonstrated the most robust neuroprotective and antioxidant activity, more pronounced than the reference melatonin and rasagiline. Statistically significant MAO-B inhibitory effects were exerted by some of the compounds where again the catecholic compound **3h** was the most potent inhibitor similar to selegiline and rasagiline. The most potent antioxidant effect in the ferrous iron induced lipid peroxidation assay was observed for the three catechols—**3h** and **3j**, **3q**. The catecholic compound **3h** showed scavenging capability against superoxide radicals and antioxidant effect in the iron/deoxyribose system. The study outlines a perspective multifunctional compound with the best safety profile, neuroprotective, antioxidant and MAO-B inhibiting properties.

## 1. Introduction

Parkinson’s disease (PD) is a major global health concern, being the second most prevalent neurodegenerative disorder after Alzheimer’s disease, affecting more than 10 million people worldwide. As the incidence of PD rises significantly with age and the human life expectancy is increasing, a dramatic rise of PD prevalence is foreseen and expected to double by 2050, reinforcing the need for new and more effective therapies [1]. Unfortunately, compared with the number of people developing PD, the research progress is extremely insufficient. PD is an incurable and progressive neurodegenerative disorder resulting in dopamine (DA) deficiency caused by the progressive death of dopaminergic neurons in the substantia nigra pars compacta, a region of the brain that controls motor activity by projecting dopaminergic axons to the striatum [2,3,4]. Despite the years of extensive research and considerable advances made towards better understanding of PD, the underlying cause still remains unknown and the applied treatment is predominantly focused on symptomatic relief with drugs. An example is the traditional levodopa replacement therapy, which allows effective treatment of PD symptoms for several years after diagnosis, but it cannot prevent the disease progression [5]. There is evidence implicating oxidative stress due to the formation of reactive oxygen species (ROS) as a key factor in the complex degenerating cascade underlying the dopaminergic neurodegeneration [6]. The substantia nigra of PD patients has shown elevated levels of oxidized proteins, DNA [7], and lipids [8]. Biochemical alterations in the substantia nigra and the frontal cortex in preclinical PD include altered oxidative stress responses, such as reduced glutathione levels, increased neuroketals and lipoxidative damage of a-synuclein, indicating that oxidative damage occurs at very early stages preceding the formation of Lewy bodies [9]. Thus, the abnormal ROS activity is more likely among the primary causes of neurodegeneration than a secondary response. Due to the presence of ROS-generating enzymes such as tyrosine hydroxylase and monoamine oxidase (MAO), the dopaminergic (DAergic) neurons are particularly prone to oxidative stress. In addition, the nigral DAergic neurons contain iron, which catalyzes the Fenton reaction causing the formation of superoxide radicals and hydrogen peroxide [10]. The inhibition of MAO-B, besides prolonging the half-life of DA leading to extended neurotransmission for relieving the motor symptoms, also prevents the further MAO-B-mediated oxidative damages during DA degradation [11,12] and decreases the parkinsonian symptoms [13,14].

Since clinical therapy capable of stopping the neurodegenerative process is not available, a neuroprotective therapy aimed at modifying the etiopathogenesis, and thus slowing down the progression, is important for more targeted therapeutic treatment [15]. Agents targeting oxidative stress are prime candidates for neuroprotection. Besides its potent MAO-B inhibiting activity, the first-line drug rasagiline also exhibits neuroprotective activity in vitro and in vivo. Rasagiline has proven effective in protecting against many toxins including 6-hydroxydopamine (6-OHDA), 1-methyl-4-phenyl-1,2,3,6-tetrahydropyridine (MPTP), and amyloid beta [16]. However, rasagiline has been related to adverse side effects [17]. Furthermore, due to the complex pathophysiology and etiology of the PD that includes a cascade of neurotoxic molecular events, the so far applied ‘one-target’ approach has demonstrated incapability to correspond to the multifactorial nature of PD. The extensive search for effective treatment has ultimately led to a new paradigm shift in the drug development. The alternative and promising approach for drug design going beyond the ‘one-molecule, one-target’ paradigm is the Multi-Target Drug Ligands (MTDL) strategy. Over the last decade, it has been successfully utilized for the development of a growing number of new drugs such as safinamide, a reversable MAO-B inhibitor (Figure 1), which was approved as an add-on treatment for PD [18]. Such drugs target an array of pathological pathways, each of which is believed to contribute to the cascades that ultimately lead to neuronal cell death. Series of methoxy and hydroxy substituted benzylideneindanone derivatives have shown multi-target activity among which the catecholic 5,6-dihydroxy compound had the highest ability to inhibit Aβ1−42 aggregation (80.1%), and MAO-B (IC_50_ = 7.5 μM), and it was also an excellent antioxidant and metal chelator [19]. The benzimidazole hydrazone derivative has shown potent and selective MAO-B inhibiting activity [20]. Recently, it was found that the presence of a catechol group at positions 7 and 8 of thiophenylcoumarins exponentially increases the antioxidant potential and the neuroprotective properties [21].

Accordingly, our research is based on the hypothesis that by eliminating major sources of oxidative stress and providing a simultaneous neuroprotection, it could be possible to delay the progression of PD. With this aim we synthesized a new class of potential MTDLs with neuroprotective, MAO-B inhibiting, and antioxidant activity. Benzimidazole-2-thione is an important fragment in medicinal chemistry that is present in many biologically active compounds including antihelmintic [22], antiviral [23], anti-inflammatory [24], anti-anticonvulsant [25] and urease inhibiting activity [26]. Furthermore, the scaffold provides convenient possibilities for structural modifications and incorporation of desired pharmacophores, and for these reasons it was selected for the creation of hybrid entities with multiple activities bearing the potential to synergistically modulate the pathological PD cascade. The compounds were designed to carry few selected privileged pharmacophores, including hydrazone moiety, which has shown in many previous studies to contribute to the MAO-B inhibiting activity [27,28,29] as well as the antioxidant activity [30]. The combination with a hydroxyl group is expected to enhance the antioxidant effects. The second pharmacophore of choice is the o-dihydroxy group, as in our previous studies catechols showed one of the highest MAO-B and neuroprotective activity [31,32]. The simultaneous presence of methoxy and hydroxy groups has shown beneficial effects in terms of pronounced neuroprotection and MAO-B on vanillin analogue [31], hence such derivatives were also synthesized and the influence of the different substituents in the arylhydrazone part examined. In our previous work we studied the in vitro protective effects of few 1,3-disubstituted benzimidazole-2-thione arylhydrazone hybrids on 6-OHDA-induced oxidative stress in rat brain synaptosomes where the vanillin derivative was distinguished as the most potent, revealing a concentration-dependent neuroprotective and antioxidant properties similar to melatonin [31,33]. Hence, the hypothesis is that the incorporation of hydroxy and methoxy groups in the arylhydrazone moiety would increase the neuroprotective activity. The compounds did not exhibit any synaptosomal toxicity at all the tested concentrations and preserved to a significant extent the synaptosomal viability and the GSH levels. Additionally, they had the ability to decrease the chemiluminescent scavenging index. All that suggests they could directly scavenge ROS and stabilize the cell membranes against free radical damage and thus bear a promising potential as therapeutic neuroprotective agents. In another study on analogous monosubstituted benzimidazole hybrids with MAO-B inhibiting activity, we outlined as a leading structure the derivative containing a 2-hydroxy-4-methoxy residue since the exhibited neuroprotective properties were stronger than the ones of melatonin and rasagiline [32]. The obtained results motivated the synthesis of a broader series of variably methoxy and hydroxy substituted hybrid molecules with improved properties, and the investigation of their potential as a new scaffold for the discovery of multifunctional drugs for neurodegenerative diseases seems promising.

## 2. Materials and Methods

### 2.1. Chemistry

All synthetic chemicals and reagents were obtained from Sigma-Aldrich (Darmstadt, Germany) and Alfa Aesar (Kandel, Germany). The SH-SY5Y neuroblastoma cell line was obtained from the European Collection of Cell Cultures (ECACC, Salisbury, UK). The cell culture medium Roswell Park Memorial Institute (RPMI) 1640 Medium, Foetal Bovine Serum 10%, L-glutamine, trypsin, 3-(4,5-di-methylthiazol-2-yl)-2,5-diphenyltetrazolium bromide (MTT), dimethylsulfoxide (DMSO), and hydrogen peroxide solution 30% (*w*/*w*) were obtained from Sigma-Aldrich (Taufkirchen, Germany); HEPES (Sigma Aldrich, Germany), Sucrose (Sigma Aldrich, Germany), NaCl (Merck, Germany), KCl (Merck), D-glucose (Merck), NaH_2_PO_4_ (Scharlau Chemie SA, Spain), CaCl_2_·2H_2_O (Merck), MgCl_2_x2H_2_O (Sigma), Percoll (Sigma), trichloroacetic acid (TCA) (Valerus, Bulgaria), 2,20-dinitro-5,50-dithiodibenzoic acid (DTNB) (Merck), human recombinant MAOB enzyme (hMAOB) kit (Invitrogen, USA), and Thyramine HCl (Sigma) were also used.

Melting points (mp) were determined using a Büchi B-540 instrument and are uncorrected. IR spectra were recorded on a Bruker Tensor 27 spectrometer in ATR mode with a diamond ATR accessory, with 64 scans at 2 cm^−1^ resolution. ^1^H and ^13^C NMR spectra were recorded on a Bruker Avance II+ 600 MHz NMR instrument. The spectra are referred to the solvent signal. Chemical shifts are expressed in ppm and coupling constants in Hz. The reactions have been monitored by thin layer chromatography performed on Merck pre-coated plates (silica gel. 60 F254, 0.25 mm) and visualized by fluorescence quenching under UV light (254 nm).

### 2.2. General Procedure for the Synthesis of Compounds ***3**–**19***

The respectively substituted hydroxy and methoxy benzaldehydes (2.0 equiv) were added to a solution of the hydrazide 2 (0.1 g, 1.0 equiv) in absolute ethanol and the solution was refluxed for approximately 2 h. The progress of the reaction was monitored using TLC (benzene:methanol = 4:1). The precipitated product was filtered and washed with ethanol. The compounds were recrystallized with ethanol and the purity was confirmed by TLC, IR, and ^1^H NMR spectroscopy.

#### 2.2.1. 3,3′-(2-Thioxo-1H-benzo[d]imidazole-1,3(2H)-diyl)bis(N′-benzylidene)propanehydrazide) (**3a**)

Yield 90%, Mp 246–248 °C, IR (ν_max_/cm^−1^) 3187 (νN-H); 2966 ν_as_CH_2_); 2925 (ν_s_CH_2_); 1675 (νC=O) amide I; 1610 (νC=N); 1558 (δN-H); 1205 (νC=S). ^1^H NMR (600 MHz, DMSO-*d*_6_) δ, ppm: δ 11.40–11.48 (d, *J* = 2H, NH), 7.89–8.09 (m, 2H, CH), 7.24–7.66 (m, 14H, Ar-H), 4.48–4.60 (m, 4H, CH_2_), 3.05–3.13 (m, 2H, CH_2_), 2.73–2.77 (m, 2H, CH_2_).^13^C NMR (151 MHz, DMSO-*d*_6_) δ, ppm:172.46, 168.53, 166.61, 165.08, 146.83, 143.68, 134.64, 134.47, 131.87, 130.48, 130.22, 129.26, 129.23, 129.20, 127.51, 127.19, 123.24, 110.47, 110.14, 32.85, 31.05.

#### 2.2.2. 3,3′-(2-Thioxo-1H-benzo[d]imidazole-1,3(2H)-diyl)bis(N′-(4-hydroxy-3-methoxybenzylidene)propanehydrazide) (**3****b**)

Yield 86%, Mp 229–231 °C, IR (ν_max_/cm^−1^) 3325 (νO-H); 3178 (νN-H); 2952 (ν_as_CH_3_); 2840 (ν_s_CH_3_); 1655 (νC=O) amide I; 1654 (νC=N); 1559 (δN-H); 1280 (νC-O-C); 1205 (νC=S). ^1^H NMR (250 MHz, DMSO-d_6_) δ, ppm: 11.15–11.32 (m 2H, NH); 9.53–9.60 (m, 2H, OH); 7.93–7.95 (d, *J* = 10.7 Hz, 1H, CH), 7.72–7.74 (d, *J* = 10.8 Hz, 1H, CH); 7.39–7.45 (m, 2H, Ar-H); 7.20–7.25 (m, 3H, Ar-H); 7.13–7.16 (dd, *J* = 9.9, 1.9 Hz 1H, Ar-H); 6.99–7.02 (m, 1H, Ar-H); 6.91–6.97 (m, 1H, Ar-H); 6.74–6.80 (m, 2H, Ar-H); 4.44–4.56 (m, 4H, CH_2_); 3.75–3.78 (m, 6H, CH_3_); 3.02–3.09 (m, 2H, CH_2_); 2.69–2.74 (m, 2H, CH_2_).^13^C NMR (151 MHz, DMSO-*d*_6_) δ, ppm: 172.19, 168.48, 149.04, 148.44, 148.30, 144.39, 131.81, 125.89, 121.49, 115.80, 110.00, 56.09, 56.05, 55.98, 31.04, 18.95.

#### 2.2.3. 3,3′-(2-Thioxo-1H-benzo[d]imidazole-1,3(2H)-diyl)bis(N′-(3,4 dimethoxybenzylidene)propanehydrazide) (**3****c**)

Yield 87%, Mp 239–241 °C, IR (ν_max_/cm^−1^) 3188 (νN-H); 2957 ν_as_CH_3_); 2832 (ν_s_CH_3_); 1663 (νC=O) amide I; 1642 (νC=N); 1575 (δN-H); 1270 (νC-O-C); 1202 (νC=S). ^1^H NMR (250 MHz, DMSO-*d*_6_) δ, ppm: 11.09–11.43 (m, 2H, NH); 7.92–7.94 (d, *J* = 12.0 Hz, 1H, CH), 7.66–7.69 (d, *J* = 17.6 Hz, 1H, CH); 7.47–7.48 (m, 1H, Ar-H); 7.17–7.23 (m, 2H, Ar-H); 7.11–7.12 (m, 1H, Ar-H); 7.07–7.10 (m, 1H, Ar-H), 6.95–7.02 (m, 1H, Ar-H), 6.86–6.93 (m, 2H, Ar-H), 4.35–4.55 (m, 4H, CH_2_); 3.70–3.74 (m, 12H, CH_3_); 2.99–3.11 (m, 2H, CH_2_); 2.63–2.75 (m, 2H, CH_2_). ^13^C NMR (151 MHz, DMSO-*d*_6_) δ, ppm: 172.21, 168.52, 148.53, 148.50, 147.52, 144.53, 144.45, 138.08, 131.84, 124.82, 124.72, 123.27, 123.20, 104.99, 104.80, 56.53, 56.51, 56.47, 56.45, 32.81.

#### 2.2.4. 3,3′-(2-Thioxo-1H-benzo[d]imidazole-1,3(2H)-diyl)bis(N′-(4-hydroxy-3,5-dimethoxybenzylidene)propanehydrazide) (**3****d**)

Yield 85%, Mp 284–286 °C, IR (ν_max_/cm^−1^) 3324 (νO-H); 3260 (νN-H); 2940 (ν_as_CH_3_); 2844 (ν_s_CH_2_); 1670 (νC=O) amide I; 1656 (νC=N); 1556 (δ N-H); 1315 (νC-O-C); 1215 (νC=S). ^1^H NMR (250 MHz, DMSO-*d*_6_) δ, ppm: 11.23–11.31 (m, 2H, NH); 8.86 (s, 1H, OH), 8.81–8.83 (d, *J* = 5.4 Hz, 1H, OH); 7.96–7.98 (d, *J* = 5.2 Hz, 1H, CH); 7.75–7.76 (d, *J* = 3.0 Hz, 1H, CH); 7.40–7.57 (m, 2H, Ar-H); 7.23–7.26 (m, 2H, Ar-H); 6.86–6.92 (m, 4H, Ar-H); 6.92–7.26 (m, 4H, Ar-H); 4.47–6.59 (m, 4H, CH_2_); 3.76–3.80 (m, 12H, CH_3_); 3.05–3.13 (m, 2H, CH_2_); 2.72–2.78 (m, 2H, CH_2_). ^13^C NMR (151 MHz, DMSO-*d*_6_) δ, ppm: 172.19, 168.50, 166.26, 148.52, 147.51, 144.62, 138.06, 131.88, 124.81, 124.70, 123.22, 110.06, 104.98, 56.52, 56.45, 33.02, 31.14.

#### 2.2.5. 3,3′-(2-Thioxo-1H-benzo[d]imidazole-1,3(2H)-diyl)bis(N′-(2-hydroxybenzylidene)propanehydrazide) (**3****e**)

Yield 86%, Mp 272–273 °C, IR (ν_max_/cm^−1^) 3200 (νO-H) 3074 (νN-H); 2940 (ν_as_CH_2_); 2855 (ν_s_CH_2_); 1663 (νC=O) amide I; 1622 (νC=N); 1560 (δN-H); 1272 (νC-O-C); 1215 (νC=S). ^1^H NMR (600 MHz, DMSO-*d*_6_) δ 11.70–11.72 (d, *J* = 8.9 Hz, 1H, NH), 11.37–11.38 (d, *J* = 8.8 Hz, 1H, NH), 11.06 (s, 1H, OH), 10.03–10.05 (d, *J* = 9.0 Hz, 1H, OH), 8.28–8.30 (d, *J* = 8.2 Hz, 1H, CH), 8.23 (d, *J* = 3.8 Hz, 1H, CH), 7.40–7.68 (m, 4H, Ar-H), 7.11–7.35 (m, 4H, Ar-H), 6.73–6.97 (m, 4H, Ar-H), 4.44–4.75 (m, 4H, CH_2_), 2.70–3.17 (m, 4H, CH_2_). ^13^C NMR (151 MHz, DMSO-*d*_6_) δ, ppm: 172.06, 168.52, 166.48, 166.46, 157.73, 156.75, 147.21, 141.39, 131.82, 131.51, 129.76, 126.77, 123.27, 120.52, 119.80, 119.06, 116.79, 116.51, 110.42, 110.26, 32.63, 31.01.

#### 2.2.6. 3,3′-(2-Thioxo-1H-benzo[d]imidazole-1,3(2H)-diyl)bis(N′-(3-hydroxybenzylidene)propanehydrazide) (**3****f**)

Yield 90%, Mp 223–224 °C, IR (ν_max_/cm^−1^) 3204 (νO-H); 3064 (νN-H); 2957 (ν_as_CH_2_); 2832 (ν_s_CH_2_); 1647 (νC=O) amide I; 1610 (νC=N); 1551 (δN-H); 1277 (νC-O-C); 1215 (νC=S). ^1^H NMR (600 MHz, DMSO-*d*_6_) δ 11.30–11.47 (m, 2H, NH), 9.75 (s, 1H, OH), 9.66– 9.67 (d, *J* = 5.8 Hz, 1H, OH), 7.96–7.97 (d, *J* = 7.3 Hz, 1H, CH), 7.80–7.81 (d, *J* = 5.2 Hz, 1H, CH), 7.45–7.52 (m, 2H, Ar-H), 7.15–7.28 (m, 4H, Ar-H), 7.09–7.10 (m, 1H, Ar-H), 6.95–7.04 (m, 3H, Ar-H), 6.75–6.80 (m, 2H, Ar-H) 4.44–4.57 (m, 4H, CH_2_), 3.05–3.10 (m, 2H, CH_2_), 2.71–2.76 (m, 2H, CH_2_). ^13^C NMR (151 MHz, DMSO-*d*_6_) δ, ppm: 172.47, 168.42, 166.78, 157.94, 147.14, 144.16, 135.80, 135.68, 131.80, 130.38, 123.34, 119.29, 118.66, 117.90, 117.59, 113.13, 113.11, 110.43, 110.17, 31.01.

#### 2.2.7. 3,3′-(2-Thioxo-1H-benzo[d]imidazole-1,3(2H)-diyl)bis(N′-(4-hydroxybenzylidene)propanehydrazide) (**3****g**)

Yield 91%, Mp 281–283 °C, IR (ν_max_/cm^−1^) 3212 (νO-H); 3068 (νN-H); 2922 (ν_as_CH_2_); 2832 (ν_s_CH_2_); 1655 (νC=O) amide I; 1561 (νC=N); 1584 (δN-H); 1237 (νC-O-C); 1212 (νC=S). ^1^H NMR (600 MHz, DMSO-*d*_6_) δ, ppm: 11.19–11.28 (m, 2H, NH), 9.78–9.94 (m, 2H, OH), 7.97–7.97 (d, *J* = 5.7 Hz, 1H, CH), 7.77–7.80 (d, *J* = 5.4 Hz, 1H, CH), 7.45–7.53 (m, 3H, Ar-H), 7.36–7.41 (m, 3H, Ar-H), 7.22–7.26 (m, 2H, Ar-H), 6.73–6.80 (m, 4H, Ar-H), 4.48–4.58 (m, 4H, CH_2_), 3.02–3.08 (m, 2H, CH_2_), 2.54–2.73 (m, 2H, CH_2_). ^13^C NMR (151 MHz, DMSO-*d*_6_) δ, ppm: 172.07, 168.50, 166.24, 159.81, 159.56, 147.19, 144.05, 131.86, 129.28, 128.92, 125.51, 123.24, 116.11, 116.06, 110.46, 110.14, 31.06.

#### 2.2.8. 3,3′-(2-Thioxo-1H-benzo[d]imidazole-1,3(2H)-diyl)bis(N′-(2,3-dihydroxybenzylidene)propanehydrazide) (**3****h**)

Yield 87%, Mp 246–248 °C, IR (ν_max_/cm^−1^) 3192 (νO-H); 3080 (νN-H); 2957 (ν_as_CH_2_); 2832 (ν_s_CH_2_); 1672 (νC=O) amide I; 1607 (νC=N); 1564 (δN-H); 1284 (νC-O-C); 1229 (νC=S). ^1^H NMR (600 MHz, DMSO-*d*_6_) δ, ppm: 11.72–11.73 (d, *J* = 5.8 Hz, 1H, NH), 11.40–11.41 (d, *J* = 5.1 Hz, 1H, NH), 10.91–10.93 (d, *J* = 10.3 Hz, 1H, OH), 9.48–9.50 (d, *J* = 9.1 Hz, 1H, OH), 9.21–9.26 (m, 2H, OH), 8.2–8.25 (m, 2H, CH), 7.51–7.55 (m, 2H, Ar-H), 7.22–7.27 (m, 2H, Ar-H), 6.90–7.02 (m, 2H, Ar-H), 6.77–6.83 (m, 2H, Ar-H), 6.61–6.72 (m, 2H, Ar-H) 4.53–4.59 (m, 4H, CH_2_), 3.06–3.09 (m, 2H, CH_2_), 2.76–2.79 (m, 2H, CH_2_). ^13^C NMR (151 MHz, DMSO-*d*_6_) δ, ppm: 171.91, 168.51, 166.45, 166.42, 147.95, 146.38, 146.00, 145.59, 142.34, 131.85, 131.82, 123.30, 120.88, 120.34, 119.68, 119.59, 119.15, 117.75, 117.43, 116.97, 110.42, 56.50, 32.62, 30.93.

#### 2.2.9. 3,3′-(2-Thioxo-1H-benzo[d]imidazole-1,3(2H)-diyl)bis(N′-(2,4-dihydroxybenzylidene)propanehydrazide) (**3****i**)

Yield 89%, Mp 254–256 °C, IR (ν_max_/cm^−1^) 3205 (νO-H); 3068 (νN-H); 2957 ν_as_CH_2_); 2832 (ν_s_CH_2_); 1654 (νC=O) amide I; 1631 (νC=N); 1555 (δN-H); 1270 (νC-O-C). ^1^H NMR (600 MHz, DMSO-*d*_6_) δ, ppm: 11.52–11.33 (d, *J* = 5.2 Hz, 1H, NH), 11.23–11.24 (d, *J* = 6.1 Hz, 1H, OH), 11.19–11.20 (d, *J* = 3.5 Hz, 1H, NH), 9.78–10.00 (m, 3H, OH), 8.15–8.16 (d, 1H, CH), 8.10 (s, 1H, CH), 7.50–7.67 (m, 2H, Ar-H), 7.35–7.37 (dd, *J* = 8.5, 1.8 Hz, 1H, Ar-H), 7.22–7.27 (m, 3H, Ar-H), 6.24–6.33 (m, 1H, Ar-H), 6.27–6.28 (m, 2H, Ar-H), 6.24–6.25 (m, 1H, Ar-H) 4.51–4.57 (m, 4H, CH_2_), 3.01–3.05 (m, 2H, CH_2_), 2.72–2.75 (m, 2H, CH_2_).^13^C NMR (151 MHz, DMSO-*d*_6_) δ, ppm: 171.53, 168.50, 166.02, 165.99, 161.11, 160.79, 159.75, 158.41, 148.18, 131.85, 131.81, 131.61, 128.57, 123.25, 111.99, 110.84, 110.42, 108.24, 108.10, 103.03, 102.77, 65.41, 32.61, 30.94.

#### 2.2.10. 3,3′-(2-Thioxo-1H-benzo[d]imidazole-1,3(2H)-diyl)bis(N′-(3,4-dihydroxybenzylidene)propanehydrazide) (**3****j**)

Yield 87%, Mp 212–214 °C, IR (ν_max_/cm^−1^) 3220 (νO-H); 3190 (νN-H); 2957 ν_as_CH_2_); 2832 (ν_s_CH_2_); 1663 (νC=O) amide I; 1631 (νC=N); 1563 (δN-H); 1283 (νC-O-C); 1220 (νC=S).^1^H NMR (600 MHz, DMSO-*d*_6_) δ, ppm: 11.23–11.24 (d, *J* = 3.6 Hz, 1H, NH), 11.18–11.19 (d, *J* = 3.8 Hz, 1H, NH), 9.36–9.38 (m, 2H, OH), 9.25 (s, 1H, OH), 9.15–9.16 (d, *J* = 3.3 Hz, 1H, OH), 7.78–7.89 (d, *J* = 4.8 Hz, 1H, CH), 7.74–7.76 (d, *J* = 2.2, 1H, CH), 7.51–7.56 (m, 2H, Ar-H), 7.23–7.28 (m, 2H, Ar-H), 7.15–7.16 (d, *J* = 1.9 Hz, 1H, Ar-H), 7.04–7.05 (d, *J* = 2.0 Hz, 1H, Ar-H), 6.71–6.87 (m, 4H, Ar-H), 4.51–4.58 (m, 4H, CH_2_), 3.02–3.08 (m, 2H, CH_2_), 2.70–2.72 (m, 2H, CH_2_).^13^C NMR (151 MHz, DMSO-*d*_6_) δ, ppm: 171.89, 168.49, 166.19, 152.60, 152.18, 148.36, 148.13, 147.35, 146.13, 146.08, 144.45, 131.87, 131.83, 126.04, 125.95, 123.29, 121.01, 120.48, 116.02, 115.96, 113.05, 112.96, 110.29, 32.88, 31.18, 31.01.

#### 2.2.11. 3,3′-(2-Thioxo-1H-benzo[d]imidazole-1,3(2H)-diyl)bis(N′-(2,3,4-trihydroxybenzylidene)propanehydrazide) (**3****k**)

Yield 82%, Mp 279–280 °C, IR (ν_max_/cm^−1^) 3316 (νO-H); 3121 (νN-H); 2966 ν_as_CH_2_); 2880 (ν_s_CH_2_); 1678 (νC=O) amide I; 1633 (νC=N); 1549 (δN-H); 1240 (νC-O-C); 1215 (νC=S). ^1^H NMR (600 MHz, DMSO-*d*_6_) δ, ppm: 11.57 (s, 1H, OH), 11.28–11.31 (m, 2H, NH), 9.61 (d, *J* = 2.8 Hz, 1H, OH), 9.46–9.50 (m, 2H, OH), 8.48–8.50 (d, *J* = 9.9 Hz, 2H, OH), 8.08–8.12 (m, 2H, CH), 7.52–7.58 (m, 2H, Ar-H), 7.23–7.28 (m, 2H, Ar-H), 6.83–6.84 (m, 2H, Ar-H), 6.72–6.74 (m, 2H, Ar-H), 4.54–4.69 (m, 4H, CH_2_), 3.02–3.05 (m, 2H, CH_2_), 1.54–2.76 (m, 2H, CH_2_).^13^C NMR (151 MHz, DMSO-*d*_6_) δ, ppm: 171.33, 168.49, 166.02, 149.23, 149.13, 148.72, 147.81, 146.97, 133.11, 131.81, 123.30, 121.49, 112.36, 111.13, 110.41, 108.04, 56.51, 32.59.

#### 2.2.12. 3,3′-(2-Thioxo-1H-benzo[d]imidazole-1,3(2H)-diyl)bis(N′-(2-hydroxy-3-methoxybenzylidene)propanehydrazide) (**3****l**)

Yield 88%, Mp 268–270 °C, IR (ν_max_/cm^−1^) 3198 (νO-H); 3061 (νN-H); 2937 (ν_as_CH_2_); 1660 (νC=O) amide I; 1642 (νC=N); 1560 (δN-H); 1253 (νC-O-C); 1215 (νC=S). ^1^H NMR (600 MHz, DMSO-*d*_6_) δ, ppm: 11.68–11.69 (d, *J* = 8.2 Hz, 1H, NH), 11.39–11.40 (d, *J* = 7.6 Hz, 1H, NH), 10.75–10.74 (d, *J* = 5.1 Hz, 1H, OH), 9.35–9.37 (d, *J* = 10.7 Hz, 1H, OH), 8.25–8.31 (m, 2H, CH), 7.49–7.55 (m, 2H, Ar-H), 7.22–7.27 (m, 2H, Ar-H), 7.15–7.18 (m, 1H, Ar-H), 7.08–7.10 (m, 1H, Ar-H), 6.99–7.01 (m, 1H, Ar-H), 6.93–6.95 (m, 1H, Ar-H), 6.81–6.84 (m, 1H, Ar-H), 6.72–6.76 (m, 1H, Ar-H), 4.52–4.59 (m, 4H, CH_2_), 3.78–3.79 (m, 6H, CH_3_), 3.05–3.10 (m, 2H, CH_2_), 2.75–2.79 (m, 2H, CH_2_). ^13^C NMR (151 MHz, DMSO-*d*_6_) δ, ppm: 172.08, 168.52, 166.44, 148.34, 147.41, 147.07, 146.27, 131.86, 123.25, 121.07, 120.91, 119.60, 119.47, 119.29, 118.14, 114.14, 113.23, 110.44, 110.25, 56.26, 32.66, 30.99.

#### 2.2.13. 3,3′-(2-Thioxo-1H-benzo[d]imidazole-1,3(2H)-diyl)bis(N′-(2-hydroxy-4-methoxybenzylidene)propanehydrazide) (**3****m**)

Yield 87%, Mp 247–249 °C, IR (ν_max_/cm^−1^) 3233 (νO-H); 3062 (νN-H); 2919 ν_as_CH_3_); 2839 (ν_s_CH_3_); 1635 (νC=O) amide I; 1609 (νC=N); 1543 (δN-H); 1281 (νC-O-C). ^1^H NMR (600 MHz, DMSO-*d*_6_) δ, ppm: 11.59–11.60 (d, *J* = 8.0 Hz, 1H, NH), 11.38–11.39 (d, *J* = 2.2 Hz, 1H, OH), 11.25–11.26 (d, *J* = 7.2 Hz, 1H, NH), 10.17–10.19 (d, *J* = 11.2 Hz, 1H, OH), 8.19–8.20 (d, *J* = 7.0 Hz, 1H, CH), 8.12–8.13 (d, *J* = 4.3 Hz, 1H, CH), 7.49–7.53 (m, 2H, Ar-H), 7.35–7.47 (m, 2H, Ar-H), 7.23–7.26 (m, 2H, Ar-H), 6.38–6.49 (m, 4H, Ar-H), 4.51–4.57 (m, 4H, CH_2_), 3.70–3.75 (m. 6H, CH_3_), 2.73–2.77 (m, 2H, CH_2_), 2.75–2.79 (m, 2H, CH_2_); ^13^C NMR (151 MHz, DMSO-*d*_6_) δ, ppm: 171.71, 168.51, 166.16, 162.48, 162.20, 159.67, 158.29, 147.84, 131.81, 131.45, 128.39, 123.30, 113.44, 112.08, 110.42, 106.88, 101.55, 101.31, 55.76, 55.61, 32.59.

#### 2.2.14. 3,3′-(2-Thioxo-1H-benzo[d]imidazole-1,3(2H)-diyl)bis(N′-(2-hydroxy-6-methoxybenzylidene)propanehydrazide) (**3****n**)

Yield 85%, Mp 272–274 °C, IR (ν_max_/cm^−1^) 3175 (νO-H); 3012 (νN-H); 2923 ν_as_CH_3_); 2850 (ν_s_CH_3_); 1651 (νC=O) amide I; 1623 (νC=N); 1573 (δN-H); 1241 (νC-O-C); ^1^H NMR (600 MHz, DMSO-d_6_) δ, ppm: 11.53–11.58 (d, *J* = 7.1 Hz, 1H, NH), 11.25–11.27 (d, J = 6.3 Hz, 1H, NH), 10.56–10.59 (d, *J* = 5.2 Hz, 1H, OH), 9.29–9.32 (d, *J* = 9.7 Hz, 1H, OH), 8.21–8.26 (m, 2H, CH), 7.38–7.44 (m, 2H, Ar-H), 7.25–7.28 (m, 2H, Ar-H), 7.12–7.17 (m, 1H, Ar-H), 7.01–7.08 (m, 1H, Ar-H), 6.97–7.01 (m, 1H, Ar-H), 6.89–6.94 (m, 1H, Ar-H), 6.74–6.82 (m, 1H, Ar-H), 6.67–6.71 (m, 1H, Ar-H), 4.49–4.57 (m, 4H, CH_2_), 3.72–3.75 (m, 6H, CH_3_), 3.04–3.09 (m, 2H, CH_2_), 2.72–2.76 (m, 2H, CH_2_). ^13^C NMR (151 MHz, DMSO-*d*_6_) δ, ppm: 172.21, 168.55, 167.42, 148.38, 146.31, 146.05, 145.87, 131.76, 123.23, 121.05, 120.86, 119.62, 119.57, 119.24, 118.12, 114.12, 112.23, 110.35, 110.28, 56.27, 32.66.

#### 2.2.15. 3,3′-(2-Thioxo-1H-benzo[d]imidazole-1,3(2H)-diyl)bis(N′-(3-hydroxy-4-methoxybenzylidene)propanehydrazide) (**3****o**)

Yield 90%, Mp 255–257 °C, IR (ν_max_/cm^−1^) (νO-H); 3074(νN-H); 2934 ν_as_CH_3_); 2839 (ν_s_CH_3_); 1663 (νC=O) amide I; 1619 (νC=N); 1579 (δN-H); 1271 (νC-O-C). ^1^H NMR (600 MHz, DMSO-*d*_6_) δ, ppm: ^1^H NMR (600 MHz, DMSO-*d*_6_) δ 11.30–11.31 (d, *J* = 5.8 Hz, 1H, NH), 11.24–11.25 (d, *J* = 6.7 Hz, 1H, NH), 9.30 (s, 1H, OH), 9.20–9.21 (d, *J* = 5.0 Hz, 1H, OH), 7.52–7.93 (d, *J* = 5.3 Hz, 1H, CH), 7.78–7.79 (d, *J* = 2.8 Hz, 1H, CH), 7.50–7.55 (m, 2H, Ar-H), 7.24–7.27 (m, 2H, Ar-H), 7.18–7.19 (m, 1H, Ar-H), 7.07–7.08 (dd, *J* = 4.9, 2.0 Hz, 1H, Ar-H), 6.89–6.99 (m, 4H, Ar-H), 4.51–4.59 (m, 4H, CH_2_), 3.76–3.78 (m, 6H, CH_3_), 3.05–3.09 (m, 2H, CH_2_), 2.71–2.74 (m, 2H, CH_2_). ^13^C NMR (151 MHz, DMSO-*d*_6_) δ, ppm: 172.04, 168.49, 166.33, 150.19, 149.97, 147.26, 147.20, 147.02, 144.09, 131.86, 131.83, 127.42, 127.34, 123.29, 120.74, 120.27, 112.67, 112.54, 112.40, 112.26, 110.48, 110.25, 56.01, 32.85, 31.01

#### 2.2.16. 3,3′-(2-Thioxo-1H-benzo[d]imidazole-1,3(2H)-diyl)bis(N′-(4-hydroxy-2-methoxybenzylidene)propanehydrazide) (**3****p**)

Yield%, Mp 205–207 °C, IR (ν_max_/cm^−1^) 3175 (νO-H); 3067 (νN-H); 2936 ν_as_CH_3_); 2840 (ν_s_CH_3_); 1656 (νC=O) amide I; 1600 (νC=N); 1508 (δN-H); 1242 (νC-O-C). ^1^H NMR (600 MHz, DMSO-*d*_6_) δ, ppm: 11.25–22.27 (d, *J* = 9.7 Hz, 1H, NH), 11.16–11.17 (d, *J* = 8.5 Hz, 1H, NH), 10.00 –10.12 (m, 1H, OH), 9.93–9.94 (d, *J* = 5.7 Hz, 1H), 8.30–8.31 (d, *J* = 6.2 Hz, 1H, CH), 8.13–8.14 (d, *J* = 5.0 Hz, 1H, CH), 7.58–7.60 (m, 1H, Ar-H), 7.47–7.52 (m, 3H, Ar-H), 7.22–7.25(m, 2H, Ar-H), 6.34–6.49 (m, 4H, Ar-H), 4.48–4.56 (m, 4H, CH_2_), 3.73–3.76 (m, 6H), 3.01–3.06 (m, 2H, CH_2_), 2.65–2.66 (m, 2H, CH_2_). ^13^C NMR (151 MHz, DMSO-*d*_6_) δ, ppm: 171.81, 167.53, 166.18, 163.54, 161.12, 159.43, 158.23, 146.82, 131.76, 131.23, 128.43, 124.40, 114.54, 111.07, 110.22, 107.64, 101.45, 101.23, 56.66, 56.62, 32.62.

#### 2.2.17. 3,3′-(2-Thioxo-1H-benzo[d]imidazole-1,3(2H)-diyl)bis(N′-(3,4-hydroxy-5-methoxybenzylidene)propanehydrazide) (**3****q**)

Yield 87%, Mp 174–176 °C, IR (ν_max_/cm^−1^) 3201 (νO-H); 3009 (νN-H); 2965 ν_as_CH_3_); 2858 (ν_s_CH_3_); 1671 (νC=O) amide I; 1607 (νC=N); 1538 (δN-H); 1293 (νC-O-C). ^1^H NMR (250 MHz, DMSO-*d*_6_) δ, ppm: δ 11.32–11.33 (d, *J* = 5.4 Hz, 1H, NH), 11.21–11.24 (d, *J* = 6.7 Hz, 1H, NH), 9.38–9.40 (m, 2H, OH), 9.30 (s, 1H, OH), 9.17–9.18 (d, *J* = 3.3 Hz, 1H, OH), 7.94–7.96 (d, *J* = 5.1 Hz, 1H, CH); 7.75–7.76 (d, *J* = 3.0 Hz, 1H, CH); 7.40–7.57 (m, 2H, Ar-H); 7.23–7.26 (m, 2H, Ar-H); 6.86–6.92 (m, 4H, Ar-H); 6.92–7.26 (m, 4H, Ar-H); 4.47–4.59 (m, 4H, CH_2_); 3.76–3.80 (m, 6H, CH_3_); 3.02–3.11 (m, 2H, CH_2_); 2.68–2.73 (m, 2H, CH_2_). ^13^C NMR (151 MHz, DMSO-*d*_6_) 172.39, 168.51, 166.28, 148.52, 149.96, 147.50, 144.49, 138.76, 131.88, 124.45, 124.38, 123.22, 110.46, 104.78, 56.54, 56.45, 33.04, 31.05.

### 2.3. Cell Culture and Cultivation

Human neuroblastoma cell line SH-SY5Y was obtained from the European Collection of Cell Cultures (ECACC, Salisbury, UK). The cells were maintained in 75 cm^2^ flasks at 37 °C in a humidified atmosphere with 5% CO_2_ and cultured in RPMI 1640 medium, supplemented with 10% fetal bovine serum and 2 mM L-glutamine.

### 2.4. MTT-Dye Reduction Assay

MTT (Sigma-Aldrich Chemie GmbH, Schnelldorf, Germany) assay is based on interaction between tetrazolium dye 3-(4,5-dimethylthiazol-2-yl)-2,5-diphenyltetrazolium bromide and succinic dehydrogenase in the living cells’ mitochondria. In this reaction, the formed purple formazan crystals indicate the presence of viable cells. SH-SY5Y cells were seeded in 96-well microplates at a density 2.5 × 10^4^ cells/well and allowed to attach to the well surface at 37 °C in a humidified atmosphere with 5% CO_2_ (24 h). Different concentrations of compounds (1, 50, 100, 250, 500, and 1000 µM) were added to cells and incubated for 24 h. For each concentration, a set of at least 8 wells were used. After the treatment, the solution in each well was substituted with MTT (3-(4,5-dimethylthiazol-2-yl)-2,5-diphenyltetrazolium bromide) solution (0.5 mg/mL in culture medium). The microplates were further incubated for 3 h at 37 °C and the obtained formazan crystals were dissolved by 100 µL/well of DMSO. The absorbance was measured in a multiplate reader Synergy 2 (BioTek Instruments, Inc, Highland Park, Winooski, VT, USA) at 570 nm (690 nm for background absorbance) [34].

### 2.5. H_2_O_2_-Induced Oxidative Stress Model In Vitro

SH-SY5Y cells were seeded in 96-well plates at a density of 3.5 × 10^4^/well and allowed to attach at the wells’ bottom overnight. After 24 h, the cell medium was aspirated and they were treated with solutions of the compounds (final concentrations 1, 10, 50 µM) for 90 min before H_2_O_2_ exposure. Afterwards, the SH-SY5Y cells were washed with phosphate-buffered saline (PBS) to remove the extracellular amount of the test compounds. Thereafter, damage on cells was achieved by hydrogen peroxide treatment (H_2_O_2_, 1 mM) in PBS for 15 min. Then, the contents of all wells were changed with culture medium. After 24 h, the amount of attached viable cells was evaluated by MTT assay. Negative controls (cells without hydrogen peroxide treatment) were considered as 100% protection and hydrogen peroxide-treated cells as 0% protection. Rasagiline and melatonin (Sigma-Aldrich Chemie GmbH) were used as reference compounds because of their well-established neuroprotective activity [35,36].

### 2.6. Measurement of Monoamine Oxidase B Enzyme Activity

The effects of the test compounds on human recombinant MAO-B (hMAO-B) enzyme were studied by the Amplex UltraRed reagent (Sigma-Aldrich Chemie GmbH) fluorometric method [37] with small modifications [38]. The working solutions of the compounds, reagents, hMAO-B, and reaction buffers were prepared according to the producer’s instructions. The tyramine hydrochloride was used as a substrate. The reaction mixture contained a stock solution of Amplex Red (10 mM in dimethyl sulfoxide), the reaction buffer (0.05 M sodium phosphate) at pH 7.4, and the storage solution of horseradish peroxidase (10 U/mL). There were pure MAO-B working solutions in the reaction buffer and MAO-B working solution containing hydrogen peroxide as controls. The final concentration of the test compounds in the well was 1 μM. The compounds together with hMAO-B were embedded in a 96-well plate (8 wells for each compound), and then the plate was incubated for 30 min (dark, at 37 °C). Thereafter, the reaction was initiated by 50 μL Mix Solution: Amplex^®^Red reagent, horseradish peroxidase, and tyramine as an enzyme substrate in the reaction buffer. The fluorescence was measured every 30 min at 0, 30, 60, 90, 120, and 150 min), in dark, while shaking the reaction mixture at constant temperature of 37 °C. The fluorimetric measurements were assessed using a Synergy 2 Microplate Reader.

### 2.7. Animals

Male Wistar rats (3 months old, body weight 200–250 g were used) were purchased from the National Breeding Center, Sofia, Bulgaria. Food and water were provided ad libitum. At least seven days of acclimatization were allowed before the study. The animals were anesthetized with ether, and perfused through the left ventricle with Tris-buffered saline, prior to decapitation. The brains were quickly removed and rinsed with the perfusion buffer and used for the preparation of synaptosomes and microsomes. All further operations were carried out at 4 °C. The animals’ health was regularly monitored by a veterinary physician. All performed procedures were approved by the Institutional Animal Care Committee and were carried out according to Ordinance No. 15/2006 for humane treatment of experimental animals (vivarium certificate of registration of farm No. 0072/01.08.2007). The experiments on isolated rat brain synaptosomes were performed according to ethical approval № 190 from the Bulgarian Agency for Food Safety. This ethical approval is valid until 6 February 2023.

### 2.8. Isolation and Incubation of Synaptosomes

Synaptosomes were prepared from brains of adult male Wistar rats. The brains were homogenized in 10 volume of cold buffer 1, containing 5 mM HEPES and 0.32 M sucrose (pH = 7.4). The brain homogenate was centrifuged twice at 1000× *g* for 5 min at 4 °C. The supernatant was collected and centrifuged 3 times at 10,000× *g* for 20 min at 4 °C. The pellet was re-suspended in ice-cold buffer 1. The synaptosomes were isolated by using Percoll reagent to prepare the gradient. Synaptosomes were re-suspended and incubated in buffer 2, containing 290 mM NaCl, 0.95 mM MgCl_2_ × 6H_2_O, 10 mM KCl, 2.4 mM CaCl_2_ × H_2_O, 2.1 mM NaH_2_PO_4_, 44 mM HEPES, and 13 mM D-glucose. Incubations were performed in a 5% CO_2_ + 95% O_2_ atmosphere [39]. The content of synaptosomal protein was determined according to the method of Lowry et al. using bovine serum albumin as a standard [40].

### 2.9. Synaptosomes Viability Assay

Synaptosomal viability was measured by (3-(4,5)-dimethylthiazol-2-yl)-2,5-diphenyl tetrazolium bromide)-test, described by Mungarro-Menchaca et al. [41] The synaptosomes were incubated with the compounds and then treated with (3-(4,5)-dimethylthiazol-2-yl)-2,5-diphenyl tetrazolium bromide) solution (0.5 mg/mL) for 1 h at 37 °C. After incubation they were centrifuged at 15,000× *g* for 1 min. The formed formazan crystals were dissolved in DMSO (Dimethyl sulfoxide). The extinction was measured spectrophotometrically at λ = 580 nm.

### 2.10. Glutathione (GHS) Assay in Isolated Synaptosomes

Following treatment with the tested compounds, the glutathione levels were determined using Ellman reagent (DTNB), which forms color complexes with the SH group at pH = 8, with maximum absorbance at 412 nm [42]. The synaptosomes were centrifuged at 500× *g* for 1 min and the sediment was used to measure GSH level. The sediment was precipitated with 5% TCA, then centrifuged for 10 min at 4000× *g* and the GSH level in the supernatant was measured using DTNB spectrophotometrically at 412 nm. The biochemical parameters were determined by spectrophotometric methods using a Spectro UV-VIS Split spectrophotometer (LaboMed.Inc, Los Angeles, CA, USA).

### 2.11. Model of 6-Hydroxy Dopamine-Induced Neurotoxicity in Isolated Rat Synaptosomes

The metabolism and oxidation of dopamine cause reactive oxygen radicals and reactive quinones to be produced, which could lead to neurotoxicity and neurodegeneration [43]. The synaptosomes were incubated with the compounds (10 µM) for 30 min and then 6-hydroxydopamnine (150 mmol/L) was used to induce neurotoxicity. The used concentration of 10 µM was chosen in accordance with the results from the H_2_O_2_-induced oxidative stress assay on SH-SY5Y cells. After 1 h, they were centrifuged on a microcentrifuge for 1 min at 15,000 rpm. The precipitates were gently mixed with buffer B and glucose and centrifuged for a second time at 15,000 rpm for 1 min. After the second wash, buffer B and glucose were added to the precipitates.

### 2.12. Statistical Analysis

Differences between the means among the tested series, treated with a corresponding concentration, were carried out by ANOVA one-way analysis of variance with the Dunnett post-hoc test. Differences were accepted to be significant when *p* < 0.05; *p* < 0.01; *p* < 0.001. All statistical analysis was carried out on the Graph Pad 6 software (GraphPad Software, Inc., La Jolla, CA, USA).

### 2.13. Iron Induced Oxidative Damage

The classical method of determining the TBA-reactive substances at 532 nm as described by Asakawa and Matsushita has been used [44,45,46]. Two groups of measurement compositions named controls and samples were prepared in phosphate buffer (K_2_HPO_4_/KH_2_PO_4_, pH 7.4) for each of the used substrates. Both of them contained equivalent concentrations of lecithin (1 mg/mL) or 2-deoxyribose (0.5 mmol/L) and FeCl_2_ (0.1 mmol/L). In the controls, the tested hydrazones have been omitted and in the samples the effect of the newly designed compounds at the concentrations indicated below the figures has been determined. The obtained samples were incubated at 37 °C for 30 min. The reaction was stopped, and after adding 0.5 mL 2.8% trichloroacetic acid and 0.5 mL of thiobarbituric acid, the samples were heated in boiling water for 20 min. The lecithin containing samples were centrifuged at 3000 rpm for 20 min and the supernatant was aspired out. The results have been presented as a percentage of the control sample, treated only with ferrous iron where we expected maximum oxidative damage under the described conditions.

### 2.14. Superoxide Scavenging Properties

Superoxide radicals scavenging activity (enzymatic assay) [47]: The assay comprises the preparation of 1 mL PBS samples containing the following substances: 1 mmol/l xanthine; 20 µL xanthine oxidase (100 IU/L); 0.04 mmol/L NBT; either the tested hydrazone or buffer for the controls. The samples were incubated at 37 °C and the absorbance was measured at 560 nm.

Superoxide radicals scavenging activity (non-enzymatic assay) [48]: 1 mL PBS, pH 7.4 contains 0.04 mmol/L NBT and the tested hydrazone in the concentrations indicated on the figures. No drug was added to the control samples. One minute after the addition of 20 μL of KO_2_ dissolved in DMSO, the absorbance at 560 nm was measured.

The results from both assays have been presented as a percentage of the control sample.

### 2.15. Molecular Docking Study of the Interactions with MAO-B

Ligands were protonated according to their protonation state at 7.4 pH and conformation library for the docking study was generated using LowModeMD methodology with AMBER10EHT force field and energy window for collection of conformations 7 kcal/mol from the lowest energy conformation. Docking was performed by the Molecular Operating Environment (MOE) 2016 software package [49]. Our model was built upon the structure of a Human monoamine oxidase B in complex with zonisamide obtained by XRD and with an overall high resolution of 1.8 Å and lack of missing residues inside the chains, PDB number 3PO7 [50]. The structure was protonated (pH 7.0, 300 K, Salt 0.1 M/L) using the 3d protonation algorithm implemented in the MOE package. Bearing in mind that the ligands might exist in different conformations, several initial structures were included in the docking study—E, s-cis, E, s-trans etc., with or without the formation of an intramolecular hydrogen bond. All conformations of all ligands were docked in the MAO B pocket using the Triangle Matcher algorithm for the initial placement of the structures, which returns up to 1 × 10^6^ poses of the ligand inside a pocket. These poses were scored by the London dG [MOE] function which estimates the free energy of the binding of the ligand from a given pose and consists of terms that estimate average gain/loss of rotational and translational entropy and loss of flexibility of the ligand. It measures the geometric imperfections of the hydrogen bonds and the desolvation energy of atoms. The best 50 poses for every ligand were further optimized with the Induced Fit methodology, using the AMBER10ETH force field/Born solvation model and optimization cutoff of 6A from the ligand. The GBVI/WSA dG [MOE] was used as rescoring function and the best 30 poses were collected for further analysis.

## 3. Results and Discussion

### 3.1. Synthesis of the Target Compounds

The synthesis of compounds **1** and **2** with the detailed experimental procedures and IR and NMR spectral data were reported in our previous studies [51]. The synthesis of the benzimidazole-aldehyde hybrids was carried out as shown in Figure 2. The ester derivative **1** was synthesized by the method previously developed by us, using Michael addition of the starting benzimidazole-2-thione to methyl acrylate in DMF medium [51]. Condensation of **1** with hydrazine hydrate in ethanol led to compound **2**, which was further refluxed in absolute ethanol with the respective methoxy and hydroxy-substituted benzaldehydes for obtaining the target 17 compounds [31].

The structures of the new compounds were confirmed by IR, ^1^H NMR, and ^13^C NMR spectra. In the IR spectra, the most characteristic bands are for the stretching vibrations of the OH group (3400–3200 cm^−1^); the stretching N-H vibrations (at ca. 3150, 3060 cm^−1^); and the amide I (νC=O) band at ca. 1660 cm^−1^. In the ^1^H NMR spectra, particularly meaningful are the characteristic N-CH_2_CH_2_CO signals (multiplets), shifted downfield due to the deshielding effect of both N-atoms and the CO groups. The chemical shift values for N-CH_2_ vary in the range from 4.44 to 4.75 ppm, while those for the CH_2_-CO-groups were found from 2.72 to 3.10 ppm appearing as two separate multiplets. Splitting was observed also for the signals of the azomethyne protons around 8 ppm and the NH groups around 11 ppm (Appendix A). The integral areas of the splitted signals follow the same ratios. In our previous studies [33] of the arylhydrazones of 1,3-disbustituted benzimidazole-2-thione, it has been demonstrated by computational methods that the compounds might exist in several confomers due to the rotation around the methylene groups and the N-N bond as well as the E or Z configuration of the azomethyne bond. It has been shown also that the conformers arising from the C-C single bonds rotation are characterized by small energy differences (0.5–3 kJ·mol^−1^), while those arising from the rotation around the N-N bond have energy differences around 8 kJ·mol^−1^ [33]. Therefore, in the current case too, it is assumed that the appearance of multiple sets of signals is connected to the presence of conformers with different orientation of the substituents around the N-N bonds. In the ^13^C NMR spectra, the signals for the C-atoms of the thione group appeared around 172 ppm, while those from the carbonyl groups—in the interval 166–168 ppm. The azomethyne C-atoms resonate at ca. 143–146 ppm. The signals for the aromatic C-atoms from the phenyl rings and the benzimidazole heterocycle are within the interval 110–130 ppm. The signals of the methylene groups were found at ca. 30–33 ppm. The signals for the C-atoms bonded to the hydroxyl groups appeared from 146 to 160 ppm depending on the position. The methoxy groups gave signals at ca. 55 ppm.

### 3.2. Neurotoxicity Profile and Neuroprotection Evaluation

#### 3.2.1. In Vitro Toxicity Evaluation on SH-SY5Y Cells

An important step in the development of new pharmacologically active compounds is the establishment of their safety profile. An in vitro toxicity evaluation was carried out on the neuroblastoma cell line SH-SY5Y, which is a well-established model in neuroscience [52,53,54]. The neuroblastoma SH-SY5Y are human derived cells with relatively homogeneous neuroblast-like structure. The cells express a dopamine-β-hydroxylase and tyrosine hydroxylase enzyme activity, specific types of human proteins and protein isoforms [55]. The possible neurotoxic effects of the potential drug candidates were measured by cell viability assay (MTT-test). The IC_50_ values were used as a robust parameter for comparison. The test compounds were dissolved in DMSO and serially diluted in cell culture medium to achieve final concentrations ranging from 1 to 500 µM; the culture medium containing DMSO in the corresponding concentration was used as untreated controls. A concentration dependent effect on cell viability was observed after 24 h incubation. The calculated IC_50_ values were in the range of 65.30–301.10 µM (Table 1).

It should be noted that compounds **3d**, **3h**, **3i**, **3n**, and **3p** did not show any significant cytotoxic effects—estimated IC_50_ values > 250 µM, followed by compounds **3e**, **3l**, and **3m** with only mild effect—estimated IC_50_ values > 200 µM, respectively. These two groups of compounds were least toxic and thus the most perspective from a safety point of view. Some of the compounds (i.e., **3f**, **3g**, **3k**, **3o**, **3q**) showed a more pronounced toxicity effect. Their calculated IC_50_ values were in the range between 65.30–77.79 µM.

#### 3.2.2. Neuroprotective Effects in H_2_O_2_-Induced Oxidative Stress on SH-SY5Y Cells In Vitro

The potential neuroprotective effects of the safest compounds were evaluated in a model of H_2_O_2_-induced oxidative stress in neuronal SH-SY5Y cells (Figure 1). The toxic mechanisms of hydrogen peroxide include the generation of reactive hydroxyl radicals and formation of products which could damage cellular lipids, proteins and DNA. The cells were pre-incubated (90 min) with the test compounds **3d**, **3e**, **3h**, **3i**, **3m**, **3n**, **3p** (1, 10, 50 µM) and then treated with H_2_O_2_ (1 mM, 15 min). The increase in cell viability compared to the H_2_O_2_-treated group was accepted for attendance of protective effect. For comparison, the effects of two compounds with well established antioxidant activity—rasagiline and melatonin—were also evaluated.

The pre-incubation of SH-SY5Y cells with melatonin and rasagiline showed protective effects against oxidative damage (Figure 2). Melatonin treatment (1 µM, 10 µM, and 50 µM) resulted in cell viability protection by 20%, 29%, and 52%, respectively (*p* < 0.001), vs. H_2_O_2_-treatment, while in the same concentrations rasagiline showed protection by 10% (*p* < 0.05), 20% and 39%, respectively (*p* < 0.001). The pre-incubation with test compounds revealed that they exhibit statistically significant protective effects. Notably, in this oxidative stress model, the neuroprotective effects of compounds **3h** and **3i** were even more pronounced than those of melatonin and rasagiline. Thus, the pretreatment of SH-SY5Y cells with compound **3h** (50 µM) and **3i** (50 µM) restored cell viability to 79% and 80%, respectively vs. H_2_O_2_-treatment (*p* < 0.001). For comparison, at the same concentration (50 µM), melatonin and rasagiline showed cell viability protection by 52% and 39% (*p* < 0.001), respectively vs. H_2_O_2_-treatment.

#### 3.2.3. Toxicity Evaluation in Isolated Rat Brain Synaptosomes

The low toxicity and substantial neuroprotective properties shown by some of the potential drug candidates in human neuroblastoma SH-SY5Y cells directed the experimental work to be elaborated on a different cellular in vitro model using freshly isolated rat brain synaptosomes for the evaluation of the protective effects and toxicity of selected promising compounds [56]. The rat brain synaptosomes represent a terminal part of the axons and contain synaptic vesicles, mitochondria, lysosomes, parts of the cell membrane, neurotransmitters, which diffuse across the synaptic cleft. Besides the presynaptic membrane, a part of postsynaptic membrane with the postsynaptic density is often attached to the nerve ending. In their experiments, Student and Edwards [57] examined the subcellular localization of the A and B forms of MAO in rat brain. Both enzymatic forms were found to be present in synaptosomal fraction. The authors conclude from the data that MAO-B activity is, indeed, associated with synaptosomes and may be non-uniformly located among different populations of synaptic endings. The functional-metabolic status of the synaptosomes was evaluated by measuring the synaptosomal viability (MTT assay) and the levels of reduced gluthathione (GSH) (Figure 2). The results from the safety evaluation of compounds **3e**, **3h**, **3i**, **3m** (1 to 250 µM) showed that they did not significantly reduce rat brain synaptosomes viability and did not change the level of GSH in concentrations < 250 µM (Figure 3). At the highest used concentration (250 µM) a relative decrease in the synaptosomal viability and the levels of reduced glutathione was observed. This effect was not significantly different from those shown by melatonin (250 µM).

Similar effects have been observed in our previous study. The compounds combining a melatonin-like structure, i.e., the benzimidazole-2-thione core, with vanilloid-like fragments, i.e., the hydroxyl/methoxy-substituted arylhydrazone moiety showed low synaptosomal toxicity in concentratons range 1–250 µM [31]. In the current case, it could also be confirmed (as it was found out for coumarins [21]), that the presence of hydroxyl groups, and in particular catechol units, may be responsible for an exponential increase in the antioxidant capacity and, indirectly, in the neuroprotective effects [21].

#### 3.2.4. Neuroprotective Effects in a Model of 6-OHDA-Induced Neurotoxicity in Rat Brain Synaptosomes

In the presented experiments, an in vitro assessment of the possible antioxidant and neuroprotective effects of the newly synthetized compounds was performed in a model of 6-OHDA induced toxicity in rat brain synaptosomes. 6-OHDA is a neurotoxicant that structurally resembles dopamine and can be used in different in vivo and in vitro models for neurotoxicity [58]. This model resembles the histological, biochemical, and physiological characteristics of neurodegenerative diseases such as Parkinson’s and Alzheimer’s disease. Neurotoxins such as MPP+ and 6-OHDA are unsuitable for systemic administration because they do not cross the blood-brain barrier. However, their uptake into brain synaptosomes is similar among primates and rodents and they are suitable in vitro models for PD. The main mechanism of 6-OHDA toxicity includes induction of oxidative stress [59]. It undergoes auto-oxidation or metabolic degradation to *p*-quinone and produces hydrogen peroxide, superoxide, and hydroxyl radicals. The process causes lipid peroxidation, protein oxidation, and DNA oxidation and finally results in oxidative stress and mitochondrial dysfunction [60]. Rat brain synaptosomes were pretreated with compounds **3e**, **3h**, **3i**, **3m**, melatonin, and rasagiline (10 μM); after that, the samples were subjected to 6-OHDA (150 mmol/L, 1 h). The protective effects were evaluated by measurement of synaptosomal viability (MTT-test) and GSH levels (Figure 3). As expected, 6-OHDA significantly reduced synaptosomal vitality and the level of GSH, respectively, by 56% and 50% compared to the untreated controls.

Rasagiline and melatonin diminished the lesions caused by the neurotoxicant 6-OHDA. Both compounds increased synaptosomal viability by 61% and 68%, respectively, and restored GSH levels, respectively by 60% and 70% vs. 6-OHDA treatment (Figure 4) Among all of the tested compounds, 10 showed the most promising protective effect similar to the reference neuroprotectors. It significantly abolished the decrease in synaptosomal viability, caused by 6-OHDA (by 71%, *** *p* < 0.001) and restored the GSH level to 62% (*** *p* < 0.001), compared to 6-OHDA treatment. Compounds **3e**, **3i**, and **3m** exhibited less pronounced protective effects.

### 3.3. Inhibitory Activity towards hMAOB and Modeling of the Interactions by Molecular Docking

Monoamine oxidase B catalyzes dopamine metabolism and oxidation leading to the formation of reactive oxygen species and reactive quinones, which provoke dopamine neurotoxicity and neurodegeneration [61]. In the next experiments, the compounds showing the best safety profile and the most promising neuroprotective activity (**3d**, **3e**, **3h**, **3i**, **3m**, **3n**, **3p**) were evaluated for their potential inhibitory activity on hMAOB. Selegiline and rasagiline as potent MAO-B inhibitors were used as standard compounds for comparison.

The results indicated that all of the tested compounds had statistically significant inhibitory effects on hMAOB activity at concentration 1 μM (Figure 4). Notably, the catecholic compound **3h** was the most potent with an hMAO-B inhibitory activity similar to selegiline and rasagiline. The current results indicate that the tested compounds effectively inhibit MAO-B at low concentration of 1 µM. This promising result should be elaborated by kinetic experiments and studies for selectivity and reversibility in order to be fully clarified the inhibition mechanism.

The interactions of the compounds with MAO-B were explored by a docking study. The best interaction energy was observed for the catecholic 2,3-dihydroxy compound **3h** and the best interaction energies of all studied compounds were found in the 5.5 kcal/mol energy window (EW), where **3h** has about 30% better interaction energy than the next compound **3p** (4-hydroxy-2-methoxy derivative). Compound **3n** (2-hidroxy-6-methoxy) has the worst predicted interaction energy. The interactions of the ligands with MAO-B are illustrated on Figure 5. The positions of the ligands in the MAO-B cavity are represented in the left part of the figure, where the molecular surface of the cavity, which is close to the ligand, is represented as a colored surface, with lipophilic part in white, hydrogen bond donor/acceptor in pink, and mild polar part in blue. The interactions of the ligand in the cavity are represented in the right part of Figure 5. All pictures use the same coloring and in all 3D structures, as well as interaction maps, the FAD moiety of MAOB is in the right part of the picture. Docked structures are represented by the balls and sticks method, while FAD is represented by sticks.

The flexible character of the hydrazone chains attached to the N-atoms of the benzimidazole core can contribute favorably to the better fitting of the ligands within the active pocket and the formation of stronger interactions. The active site pocket of MAO-B enzyme is narrow and highly lipophilic; therefore, in order to enter deeper and interact more strongly, the flexible parts of the ligands may change according to the cavity shape. In some cases, the intramolecular hydrogen bonding might be cleaved in order to form more favorable interactions with the amino acid residues of the enzyme. For example, the best binding pose of compound **3h** presented in Figure 5, which does not feature intramolecular hydrogen bonds between the hydroxyl groups or the *o*-OH groups and the N-azomethyne atoms, is more favorable than the binding poses with intramolecular hydrogen bonds by 1.073 to 1.118 kcal·mol^−1^ (Appendix A).

In all cases, the structures are placed along the cavity with one of their phenolic rings situated close to FAD and another one placed somewhat close to the entrance of the cavity. This is due to the shallow form of the active site, as it has already been mentioned by other authors [62,63]. This is in contrast to the monosubstituted benzimidazole derivatives [32], which even in their best conformations are not long enough to reach the close proximity of FAD. Current structures fill the whole active site and one of their phenolic rings is in close proximity to the flavine part of the FAD. Due to the high lipophilicity of the MAO-B active site pocket, that does not favor interactions with OH and OCH_3_ groups. The substituted phenolic rings tend to be placed close to the entrance of the channel where they can interact with the solvent or are close to the flavine ring. The arrangement of the four residues, Ile199, Phe168, Leu171, and Tyr326, that form the boundary between the two cavities of the enzyme active site, the entrance, and the substrate one [64] tend to be close to the benzimidazole ring.

### 3.4. Iron-Induced Oxidative Damage of Lecithin

Neuronal tissue is characterized by high oxygen consumption and high levels of polyunsaturated fatty acids, which are known for being extremely prone to oxidative damage [8]. Elevated levels of lipid hydroperoxides and MDA, which are both indicators of lipid peroxidation, were found in the blood of early-stage PD patients, suggesting that treatments using the current antiparkinsonian drugs should be combined with drugs targeting not only dopaminergic pathway but lipid peroxidation and lowered antioxidant levels [65]. Since the increased iron content in substantia leads to the generation of hydroxyl radicals via Fenton reaction able to attack the polyunsaturated fatty acids and form the peroxyl radical (ROO^•^) responsible for the chain reaction propagating the lipid peroxidation, the efficacy of new drug candidates in decreasing the lipid oxidative damage is an important additional beneficial feature. The obtained results indicate that all tested compounds possess the capability to influence lecithin oxidative damage by decreasing the absorbance compared to the control samples. Lower absorbance values suggest decreased generation of TBA-RS products and decreased% of the control value. The three used standard reference compounds exert different capability to decrease lecithin peroxidation (Figure 6). The absorbances of the sample containing melatonin were more than two times higher compared to that of Quercetin and Trolox. For the melatonin-containing samples, the estimated% of the control ratio decreased to 75% compared to the controls and for Trolox and Quercetin to around 25%. Despite the fact that the former two compounds have similar influence on the extent of oxidative damage, the observed effect was statistically different, and Quercetin was the better antioxidant.

The values of the sample/control absorbance ration were in the range of 40–70%. The observed properties correlate well with the types of the structural modifications and their position in the molecular structure. All compounds have lower absorbance value compared to melatonin, corresponding to statistically significant lower lecithin peroxidation. The parameter determined in the system in the presence of compound **3a** containing a residue of unsubstituted benzaldehyde was 66.52%. Comparing this value with the other compounds, we found a group of seven compounds with statistically identical effect to **3a**. These were predominantly compounds containing both OH and OCH_3_ moieties—**3l**–**3p**, the 2,4-dihydroxy hydrazone **3i**, the 2,3,4-trihydroxy hydrazone **3k**, and the veratraldehyde derivative **3c**. Another group of compounds also demonstrated statistically identical capability to decrease the extent of the observed lecithin peroxidation but corresponding to a significantly higher antioxidant effect—this was the group of the most potent protectors in the system, comprising the benzimidazole-2-thione derivative containing residues of syringaldehyde **3d**, the two catecholic compounds (2,3-dihydroxy and 3,4-dihydoxy) **3h** and **3j**, and the 3,4-dihydroxy-5-methoxy hydrazone derivative **3q**. The observed sample/control absorbance ration for all these compounds was around 40%.

### 3.5. Scavenging Capability against Superoxide Radicals and Effect on Fe Induced Degradation of Deoxyribose

The possibility to diminish the concentration of superoxide anion radicals is essential since it is related to the autoxidation process of dopamine and the formation of dopamine quinones that are able to undergo redox reactions generating highly reactive aminochrome leading to the production of superoxide and subsequently deplete the cellular NADPH. The superoxide anion radical can initiate a cascade of reactions associated with the generation of more reactive and harmful ROS via enzyme- or metal-catalysed reactions. Furthermore, an increased SOD activity is present in the late stages of PD [66,67]. The O_2_^−●^ can easily dismutate, and the obtained hydrogen peroxide is relatively stable and possesses the capability to perform facilitated diffusion across the cellular membranes. The presence of both moderately reactive ROS (O_2_^−●^ and H_2_O_2_) in combination with the increased iron content in some part of the brain tissue are favorable for the generation of the most reactive among all ROS—the hydroxyl radical [68]. For these reasons, the superoxide radical is a key target in the development of new therapeutic agents with antioxidant properties.

The potency of the compounds selected as the most promising from the cell model systems and hMAOB inhibitory studies was estimated against enzymatically (xanthine/XO system) and non-enzymatically (KO_2_/NBT) produced superoxide radicals (Figure 7). In the xanthine-xanthine oxidase system the generation of O_2_^−^^●^ is a result of an enzymatic reaction. The molecular mechanism of action of the XO enzyme is associated with catalysation of the reaction of oxidation of hypoxanthine and xanthine and generation of uric acid. During the process of reoxidation of the enzyme, the molecular oxygen acts as an electron acceptor and the formation of physiologically important ROS, such as the superoxide radical (O_2_^−^^●^) and hydrogen peroxide (H_2_O_2_), has been observed. In the potassium superoxide-containing systems, superoxide radical generation is due to the direct addition of the KO_2_ to the water sample solution. The release of a superoxide anion was estimated via NBT reduction activity, indicating detectable superoxide anion levels. The application of these two alternative model systems will afford the possibility to perform a more accurate interpretation of the obtained results. Since in the enzyme-free model system, the release of the superoxide in the small sample volume is a very fast process, there is a risk of local effects that is avoided in the xanthine-xanthine oxidase system. The use of enzyme reaction provides a relatively slow generation of the superoxide radical and steady concentration.

In the enzymatic assay, only compound 3h demonstrated scavenging activity whereas at the highest tested concentration the “% of control” was close to 90%. In the presence of compound **3e**, **3i**, and **3m** at concentrations higher than 50 µM significant increases in the absorbance of the samples compared to the controls was noticed. The effect was concentration-dependent. It was moderate in the presence of compound **3m** at the highest tested concentration of 100 µM—the “% of control” ratio was around 125%. Much more substantial increase was observed in the presence of the other two compounds **3e** and **3i**. At concentration of 100 µM the increase was respectively 1.5 and 2.5 times compared to the controls. In our previous experiments, it was proven by using enhanced chemiluminescence that analogous benzimidazole-aldehyde hybrids possess the capability to decrease the chemiluminescencent scavenging index in the system of KO_2_ generated superoxide anion radical and they acted as pronounced scavengers [31]. The reduction of NBT to formazan is an oxidation reaction of the superoxide anion radical, which is a reducing agent in the described reaction system. A possible reason for the obtained data, given the proven radical scavenging activity against the superoxide anion radical in the O_2_^●–^ containing chemiluminescent system, is the presence of strong electron donor properties in the tested compounds, combined with the lack of capability for direct interaction with NBT. In such a situation, they would act as a second reducing agent, interacting with the already released O_2_ to superoxide, which will subsequently interact with NBT.

In the xanthine/xanthine oxidase model system, the salicyl derivative **3e** and the 2-hydroxy-4-methoxy **3m** did not induce statistically significant change in the sample absorbance values in the whole tested concentration range when compared to the controls suggesting lack of effect in the system. Compounds **3h** (2,3-dihydroxy-) and **3i** (2,4-dihydroxy) decreased NBT reduction during the xanthine/xanthine oxidase assay. At the maximal tested concentration compound **3h** decreased the “% of the control” ratio to 64%. The effect of compound **3i** was moderate and only 15% decrease was observed. The observed effect could be attributed either to the capability of the compounds to capture the superoxide radical or capacity to modulate the activity of the enzyme xanthine oxidase.

The capability of the most potent compound **3h** to modulate iron induced deoxyribose degradation was estimated. Several alternative variants of the test have been proposed over the years—H_2_O_2_/Fe(III)EDTA/Ascorbic acid [69]; H_2_O_2_/Fe(III)EDTA variant; Fe(III)EDTA/Ascorbic acid variant; Fe(III)EDTA variant [70]. The described methods allow to obtain information concerning different aspects of the ROS scavenging abilities of the tested substances. The advantage is that the systems could give information concerning both—possible antioxidant and/or pro-oxidant effects of the tested compounds. The obtained results indicate decrease of the absorbance values of the samples containing **3h** compared to the controls (C) suggesting lower level of MDA-like products reactive with TBA. The observed antioxidant effect was stronger than the reference melatonin and similar to Trolox.

Another experiment was conducted with the most promising catechol compound **3h** in order to evaluate its capability to protect the deoxyribose molecules from iron induced oxidative damage (Figure 8). It denoted lack of capability to increase the induced by Fe(II) oxidative damage, better protection effect than the reference melatonin, and a similar effect to the antioxidant Trolox.

These data demonstrate an additional beneficial effect associated with the prevention of structural damage associated with changes of DNA sugar (deoxyribose) that might induce strand breaking and formation of terminal fragmented sugar residues.

## 4. Conclusions

The properties of the compounds subject of the current study surpass all our previous results on benzimidazole aldehyde hybrids as potential multi-functional compounds in terms of MAO-B inhibition and neuroprotection. The in vitro safety evaluation on SH-SY5Y cells and rat brain synaptosomes showed a strong safety profile. Moreover, compounds **3h** and **3i** possess the highest IC_50_ values, 295.90 and 301.11 µM calculated in SH-SY5Y cells, did not significantly reduce rat brain synaptosomes viability and the level of GSH in concentrations < 250 and were defined as the least toxic. The in vitro antioxidant activities and neuroprotective effects were evaluated in H_2_O_2_-induced oxidative stress on SH-SY5Y cells and in the model of 6-OHDA-induced neurotoxicity in rat brain synaptosomes and showed that compounds **3h** and **3i** demonstrated the most robust antioxidant activity. These effects were even more pronounced than those of melatonin and rasagiline. Further, our in vitro study estimated statistically significant MAO-B inhibitory effects of compounds **3d**, **3e**, **3h**, **3i**, **3m**, **3n**, **3p**. It should be noted that the catecholic compound **3h** was the most potent, inhibiting hMAOB similarly to selegiline and rasagiline. Additionally, the new benzimidazole derivative **3h** demonstrated a better safety profile, higher antioxidant activity and stronger MAOB inhibitory effect than all the other evaluated compounds and was proven as the most promising for further studies. The obtained results indicated that all tested compounds possess the capability to influence lecithin oxidative damage to a different extent depending on the types of the structural modifications. The most potent protective effect in the ferrous iron induced oxidative damage of lecithin was observed for the dihydroxy substituted compounds—**3h** and **3j**, (with 2,3-dihydroxy and 3,4-dihydoxy moiety), along the 3,4-dihydroxy-5-methoxy hydrazone **3q** and the 3,5-dimethoxy-4-hydroxy hydrazone **3d**. Moreover, hydrazone **3h** showed scavenging capability against superoxide radicals and capability to decrease deoxyribose oxidation. A more general conclusion could be made based on our up-to-date studies stating that the presence of a hydroxyl group in position 2 of the arylhydrazone fragment is essential for the simultaneous display of high neuroprotective and antioxidant properties, as well as MAO-B inhibiting properties.

## Data Availability

Data are available within the article.

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
