# Peer review of "Study on the Neuroprotective, Radical-Scavenging and MAO-B Inhibiting Properties of New Benzimidazole Arylhydrazones as Potential Multi-Target Drugs for the Treatment of Parkinson’s Disease"

_antioxidants, 2022, doi:10.3390/antiox11050884_

Round 1
Reviewer 1 Report
Review of the manuscript antioxidants-1660263, Study on the neuroprotective, radical-scavenging and MAO-B inhibiting properties of new benzimidazole arylhydrazones as potential multi-target drugs for the treatment of Parkinson’s disease.
The manuscript is a resubmission of the document antioxidants-1546900. The authors made some changes and tried to improve the paper, but some problems still remain.
The authors prepared a long response to the first review problems, but in many cases the response was directed for the review. I consider that many of these responses should be better integrated in the final paper in order for the readers to understand better the research and its limitations. See for example the docking study, the use of synaptosomes, the lack of data on inhibition profile (reversible or not), the lack of experiments on MAO-A.
Again, provide in the manuscript a rational for the use of synaptosomes. Why chose this method and why not alternative ones?
The editing needs major corrections.
Author Response
The authors thank the reviewer for his suggestions. The explanations regarding the docking study and the use of synaptosomes were already present in the manuscript. Nevertheless, more information was added. The following quote encompasses the full explanation about the synaptosomes present in the manuscript:
"The rat brain synaptosomes represent a terminal part of the axons and contain synaptic vesicles, mitochondria, lysosomes, parts of the cell membrane, neurotransmitters, which diffuse across the synaptic cleft. Besides the presynaptic membrane, a part of postsynaptic membrane with the postsynaptic density is often attached to the nerve ending. In their experiments, Student and Edwards [57] examined the subcellular localization of the A and B forms of MAO in rat brain. Both enzymatic forms were found to be present in synaptosomal fraction. The authors conclude from the data that MAO-B activity is, indeed, associated with synaptosomes and may be non-uniformly located among different populations of synaptic endings... This model resembles the histological, biochemical and physiological characteristics of neurodegenerative diseases such as Parkinson's and Alzheimer’s disease. Neurotoxins such as MPP+ and 6-OHDA are unsuitable for systemic administration because they do not cross the blood-brain barrier. However, their uptake into brain synaptosomes is similar among primates and rodents and they are suitable in vitro models for PD. "
-About the docking study:
"Our model was built upon the structure of a Human monoamine oxidase B in complex with zonisamide obtained by XRD and with an overall high resolution of 1.8 Å and lack of missing residues inside the chains, PDB number 3PO7 "...Bearing in mind that the ligands might exist in different conformations, several initial structures were included in the docking study – E, s-cis, E, s-trans etc., with or without the formation of an intramolecular hydrogen bond... The active site pocket of MAO-B enzyme is narrow and highly lipophilic, therefore, in order to enter deeper and interact more strongly the flexible parts of the ligands may change according to the cavity shape. In some cases the intramolecular hydrogen bonding might be cleaved in order to form more favorable interactions with the amino acid residues of the enzyme.
-Regarding the lack of data on the inhibition profile was added the following sentence:
"The current results indicate that the tested compounds inhibit effectively MAO-B at low concentration of 1 µM. This promising result should be elaborated by kinetic experiments and studies for selectivity and reversibility in order to be fully clarified the inhibition mechanism."
-The text was additionally edited. The authors apologize for overlooking such mistakes.
Reviewer 2 Report
The manuscript has been improved.
The following minor amendments should be introduced:
1.The improper term "molecular damage" is still present in the title of subsection 2.12.
2. line 761 - Fig.6 is discussed here not Fig.1
3. lines 913-915 - the second part of the sentence is too detailed. It should be changed to "...and capability to decrease deoxyribose oxidation". The same refers to the abstract (lines 31-32)
Author Response
The authors thank again the Reviewer and apologize for the minor mistakes. The text was edited accordingly.
Reviewer 3 Report
Summary of Paper:
The authors describe a set of new MAO-B inhibitors that provide strong neuroprotection against ROS-induced neurodegeneration. This is hugely important for neurodegenerative diseases, such as PD, as there is no current cure for them and very few, if any, treatments are focused on preventing the progression of the disease state. The authors provide a complete analysis of the compounds synthesized and their ability to provide protein to cells (in cell culture and primary neurons). The overall manuscript is interesting with potentially important information regarding disease prevention; however, there are some major issues that need to be addressed before it can be considered for publication in Antioxidants. Specifically, there were a large number of grammatical errors which made it extremely difficult to focus on the content provided herein.
Major Issues:
- There are a number of grammatical errors; this reviewer recommends that the authors have a native English speaker go through the paper to help make these major revisions. This is a major problem as it made it hard for me, the reader, to be able to focus on the content of the article. I’ve listed out some of the errors that I have found that can be quickly addressed:
- Line 19: Remove the from “and stops the ROS formation”
- Lines 20-21: Add a to “moiety of (a) new series…”
- Line 22: Change the word good from “showed a good safety profile.” Something like promising or strong would be more scientific
- Line 31: What are the absorbance values mentioned here?
- Line 41: Use the PD abbreviation for Parkinson’s
- Line 42: Remove the from “a dramatic rise of the PD prevalence…”
- Line 45: Consider changing the word choice for “the amount of research addressing the problem is extremely insufficient”. One suggestion would be to switch research to progress
- Line 50: Remove the and nowadays from “towards (the) better understanding of PD, (nowadays) the underlying cause still remains”
- Line 52: End the sentence at drugs and remove “aiming to either restore the level of dopamine in the striatum or to act on striatal postsynaptic dopamine receptors”. This sentence is way to long
- Line 56: define ROS here with oxidative stress
- Line 61: Replace pointing to indicating
- Line 65: define DAergic neurons
- Lines 73-74: Should edit it to this “aimed at modifying the etiopathogenesis, and thus slowing down the progression, is an important for more targeted therapeutic treatment.”
- Line 531: The found changes (gets smaller) and appears to stay this way until Line 604
- Line 555: What was the vehicle used to add the compounds for treatment with SH-SY5Y cells? This should be used as you control and not the cultured medium
- Line 612 – 616: The font changes to gray and needs to be switched back to black
Author Response
The authors thank the Reviewer for his attentive and constructive comments. The text was checked by a professional translator working at the European Commission. All suggested by the reviewer corrections were incorporated.
- Line 555: What was the vehicle used to add the compounds for treatment with SH-SY5Y cells? This should be used as you control and not the cultured medium
The test compounds were dissolved in DMSO and serially diluted in cell culture medium to achieve final concentrations ranging from 1 to 500 µM; the culture medium containing DMSO in the corresponding concentration was used as untreated controls.
Round 2
Reviewer 1 Report
the authors made most of the suggested changes. overall, the quality of the paper is improved, but it still needs an editing check-up and some English corrections.
Reviewer 3 Report
Authors have addressed all concerns from this reviewer
This manuscript is a resubmission of an earlier submission. The following is a list of the peer review reports and author responses from that submission.
Round 1
Reviewer 1 Report
The article antioxidants-1546900, Study on the neuroprotective, radical-scavenging and MAO-B inhibiting properties of new benzimidazole arylhydrazones as potential multi-target drugs for the treatment of Parkinson’s disease, present a relatively interesting research. The quality of the manuscript is limited by multiple mistakes and in my opinion by some faults in the experiments design. The originality of the manuscript is medium, because very similar compounds were already synthesized. The test were also relatively similar. See for example:
Neda O. Anastassova, Denista Y. Yancheva, Anelia Ts Mavrova, Magdalena S. Kondeva-Burdina, Virginia I. Tzankova, Nadya G. Hristova-Avakumova, Vera A. Hadjimitova, Design, synthesis, antioxidant properties and mechanism of action of new N,N′-disubstituted benzimidazole-2-thione hydrazone derivatives, Journal of Molecular Structure, 1165, 2018, 162-176, https://doi.org/10.1016/j.molstruc.2018.03.119.
The article does not fit very well the profile of the journal. It seems designed for a journal like Molecules.
The review will point out the major problems, but there are still minor issues that the authors should really check and correct.
The synthesis part is presented in a confusing manner. Usually the new compounds are presented as 3a-x, and not numbered separately. The authors should present how the compounds were purified.
The authors should explain the 13C-RMN values. For example C=S, has a value close to 170, the C=O has a value close to 160, and so on. The authors should also indicate if in any cases the values differ for atoms that should be identical (the molecule is no longer symmetrical).
In row 510, the authors should correct, because compound 3 is not methoxy and hydroxy-substituted. Check the chemical name of compounds 3. It is wrong.
The authors present their compounds as E isomers. They should add proofs for this assumption and also for their purity. It can be a mixture of E and Z isomers. What data proves that is only one isomer? What is the accuracy of the data?
The introduction is very long and mainly out of focus. The authors should eliminate all the unnecessary details. The section 92 to 114 should be improved by adding a scheme with the mentioned structures for the reader to understand the design of the new compounds.
The section 2.9 is not clear. The glutathione levels were measured in untreated synaptosomes, or the authors did not mentioned the treatment with the compounds? Please check.
The results are presented in a confusing order. The results should start with In vitro evaluation on SH-SY5Y cells, followed by the effects on synaptosomes and after by the other experiments. The authors should not repeat methods in the results section.
Check figure 4. The x-ordinate is not concentration. The authors should explain why the compounds were tested in different concentrations.
The docking studies may contain some errors. For example in figure 6, in the case of compound 7 it is highly probable that the 2-OH to be intramolecularly hydrogen-bonded to the nitrogen. Did the authors take this possibility in consideration? In the case of compound 10, one branch seems to be presented as E, other as Z. Please explain this. Explain also the choice of the pdb that was used.
The authors should be more careful in the conclusions. They should comment the results refereeing to the dose used. The authors should choose one dose, for example 50 μM and discuss all the effects measured at this dose. The authors should point out that the compounds can inhibit also MAO-A. The inhibition of MAO-B could be irreversible (like rasagiline). The authors performed no experiment on MAO-B to determine IC50, or the kinetics of the inhibition. Please explain in the manuscript why.
Add the ethical approval for the experiment and please explain why the rat brain synaptosomes were necessary for this research. Why the author chose this method and why not alternative ones. Please detail.
Keep the same style in all the paper. Check the whole manuscript. See just some examples: Rasagiline (R83) or Rasagilin (R84)? SH-SY5Y (R369) or SHSY5Y (R372)?
Reviewer 2 Report
file attached

Reviewer 3 Report
Study on the neuroprotective, radical-scavenging and MAO-B inhibiting properties of new benzimidazole arylhydrazones as potential multi-target drugs for the treatment of Parkinson’s disease
Synopsis: In Parkinson’s disease the therapy is symptomatic and based on providing dopamine precursor to the sufferer or inhibiting the degradation of the neurotransmitter, by blocking the activity of the monoamine oxidase (MAO) degrading enzyme. Oxidative stress is also commonly observed in pathology. The authors synthesize a series of molecules derived from 1,3-disubstituted benzimidazole-2-thiones to also add antioxidant properties to the selective MAO-B inhibiting activity of the molecule.
Critic: the article is written in very poor English, as well as having several problems in punctuation. More than once repetitions of terms were also found and, moreover, the supporting bibliography is excessively outdated.
The fact that at line 593 the insertion of a title for the table was forgotten, leaving instead the heading present in the original MDPI backbone file: "Table 1. Tables should be placed in the main text near to the first time they are cited" is indicative of a lack of attention. This can also be seen in the incorrect indication of the values in the abscissa of Figure 4 and in many captions, where the phrase "Data are presented as means from X independent experiments ± SD (n = X)" is present. Often there is no correspondence between the number of biological replicates and what is reported in brackets. Moreover, the data are redundant, since “n” and “independent experiments” are generally synonymous, unless otherwise stated by the author. A better explanation can be found here: Cumming, G., Fidler, F., & Vaux, D. L. (2007). Error bars in experimental biology. The Journal of cell biology, 177(1), 7–11. https://doi.org/10.1083/jcb.200611141
Graphic representations are difficult to interpret (especially Figure 1) and to be consulted, being generally small and blurry. Among other things, this makes the significance symbols almost impossible to be read.
The work is based on the modification of 1,3-disubstituted benzimidazole-2-thiones so, a better indication on what the molecule is, could be useful, also because the extensive form of its name is present just in abstract, being reported successively in the body of the text as “Benzimidazole hydrazone derivatives”. It is suggested to provide the article with a brief descriptive part such as the one present at the beginning of a previous work of the group: Anastassova NO, Mavrova AT, Yancheva DY, Kondeva-Burdina MS, Tzankova VI, Stoyanov SS, Shivachev BL, Nikolova RP. Hepatotoxicity and antioxidant activity of some new N,N′-disubstituted benzimidazole-2-thiones, radical scavenging mechanism and structure-activity relationship. Arab. J. Chem. 2018; 11, 353–369
The authors refer to the compounds they have generated with progressive numbers in the article, but without exhaustively explaining the difference between them or the reason for their synthesis and comparison. Moreover, these numbers appear in the abstract when the reader has no idea of their identity.
In conclusion, the paper is not considered acceptable for publication. After the review of the content, it is suggested to use a native speaker or the journal's support service for a correction of the descriptive part of the paper.